# Actively Tunable Fano Resonance Based on a Bowtie-Shaped Black Phosphorus Terahertz Sensor

**DOI:** 10.3390/nano11061442

**Published:** 2021-05-29

**Authors:** Yan Huang, Yan Liu, Yao Shao, Genquan Han, Jincheng Zhang, Yue Hao

**Affiliations:** 1Wide Bandgap Semiconductor Technology Disciplines State Key Laboratory, School of Microelectronics, Xidian University, Xi’an 710071, China; hy04131432@126.com (Y.H.); hangenquan@ieee.org (G.H.); jchzhang@xidian.edu.cn (J.Z.); yhao@xidian.edu.cn (Y.H.); 2Shanghai Energy Internet Research Institute of State, Grid251 Libing Road, Pudong New Area, Shanghai 201210, China; yaoshao@epri.sgcc.com.cn

**Keywords:** black phosphorus, Fano resonance, Terahertz

## Abstract

An ultrasensitive Terahertz (THz) sensor consisting of the sub-wavelength bowtie black phosphorus (BP) and an annular gold (Au) ring is proposed. The interference between the resonance generated by the bowtie BP and the Au ring creates a Fano-type resonance and makes ultrasensitive sensing realizable. Firstly, we demonstrate the Fano resonance of the coupled nanostructures by adjusting the geometry dimensions of the Au ring and the Fermi level of BP. Moreover, the Poynting vector distributions of the THz sensor are simulated to analyze the properties of Fano resonance. Importantly, a figure of merit (FOM) value as high as 69.3 is achieved and the proposed Fano resonance sensor shows a maximum sensitivity of 9.3 μm/RIU. Our structure can function as a facile and efficient building block of biochemical nano-sensing application based on Fano resonance at THz frequency.

## 1. Introduction

The concept of Fano resonance was proposed by the Italian physicist Ugo Fano in the framework of the quantum system [1]. It describes the asymmetric line shapes that appear in the atomic absorption spectrum. In the quantum system, Fano resonance is directly derived from the destructive interference between narrow discrete resonance and broad continuous resonance, and usually exhibits sharp asymmetric spectrum [2,3,4,5]. In fact, Fano resonance can also be understood as the coupling of two oscillators with different damping rates to produce narrow and wide spectral lines. The phase of the undamped oscillator changes π at resonance, while the phase of the strongly damped oscillator changes slowly. The phase change between them results in the asymmetry of the Fano resonance [1]. For example, hybrid graphene-metal gratings with tunable Fano resonance are proposed and theoretically investigated in the THz band [6]. It includes a graphene–metal hybrid grating on a dielectric substrate. High-ohmic loss metal gratings have a broad spectrum, while low-loss graphene gratings have narrow discrete resonances. The interaction between them produces Fano resonance and produces a transmission spectrum with sharp and asymmetrical line profiles. In the plasmon system, the formation of Fano resonance can be understood as a destructive interference when bright and dark plasmon modes interact and overlap energetically, where the continuous state corresponds to the bright mode and the discrete state corresponds to the dark mode [7,8,9,10]. The bright mode can be directly excited by incident light to resonate. It has a large radiation broadening, and is characterized by strong radiation loss and a large net dipole moment, which is easily excited by light [1]. On the contrary, the dark mode has a smaller radiation broadening, and is characterized by weak radiation loss and not easy to be directly excited by light [1]. For example, the intensity modulation and frequency modulation of Fano resonance by two coplanar and vertical nanoribbons of graphene nanodimer is proposed [11]. The Fano resonance is induced by the destructive interference between the bright dipole mode of the short nanoribbon and the dark quadrupole mode of the long nanoribbon. It is known that Fano resonance is related to the geometry of the structure and the electromagnetic parameters of the surrounding medium [2]. So, many kinds of systems have been proposed to exhibit Fano resonance, such as meta-surfaces [12], plasmonic structures [13,14], optical cavities [15], and photonic crystals [16]. For example, Fano resonance also can be generated by the coherent coupling between bright and dark plasmon patterns supported by long and short graphene nanoribbons, respectively [11]. Due to the unique characteristics of Fano resonance, there are many potential applications of Fano resonance in active plasmon switching [17,18], dual-channel switching [19], all-optical switching [20], biological and chemical sensors [21,22], surface-enhanced Raman scattering (SERS) [8], flexible meta-photonic devices [23], scalable dynamic photonic devices [24], and plasmon line shaping [25]. Especially, for its sharp asymmetric characteristics, high sensitivity, and a large figure of merit (FOM) can be easily achieved [26], so it is promising in the application of high-performance biochemical sensors. In particular, the booming development of sensors in the Terahertz (THz) frequency range requires very high sensitivity. For example, an ultrasensitive THz sensor consisting of a sub-wavelength graphene disk and an annular gold ring was demonstrated to achieve frequency sensitivity as high as 1.9082 THz/refractive index unit (RIU) [27]. The studied stacked Fano meta-structure was illustrated to achieve a record high sensitivity of 1.76×10^5^ nm/RIU and FOM of around 14.05 in the terahertz range [28]. However, due to the lack of natural materials that can strongly interact with THz waves, its implement is still a challenging task. Besides, so far, most of the Fano resonances observed in many natural nanostructures are achieved at a fixed frequency, such as dolmen structures [29], non-concentric ring or disk cavity [3], and cluster of nanospheres [30]. This problem can be solved by designing the metamaterial (MM). The MM has an artificial geometric structure, and the feature size is smaller than the wavelength of its action, so it can affect the wave. For example, the Fano resonance has been investigated based on the double ring MM structure in the THz regime, which is tunable by changing the structural parameters [31].

Interestingly, nowadays many studies have proven that plasmon can be excited in 2D materials [32]. Due to the characteristics of collective excitation of 2D materials, the plasmon confinement is much stronger than that of metal surface plasmons [33]. However, graphene has one limitation and drawback that cannot be ignored compared with BP, which is the severe scattering loss. Since the wave vectors of graphene plasmons and free-space photons do not match, the efficiency of light-excited plasmons is very low [33]. This severely limits its application on Fano resonance. BP has attracted extensive attention due to its excellent optical and electrical performance, such as stronger interlayer interaction, higher carrier mobility and tunable band gap [34,35,36,37]. Moreover, as a 2D material with direct band gap, BP has a high coupling efficiency between plasmons and phonons and low scattering loss [35]. In particular, BP is an excellent candidate to design frequency-tunable nanostructures since its Fermi level can be changed by chemical doping to further change its photoelectric properties.

In this paper, we design a tunable THz sensor with Fano resonance based on a hybrid MM structure, which is composed of the Au ring and bowtie BP structure. By changing the Fermi level of BP and the geometric parameters of the Au ring, the amplitude and resonance frequency of the Fano resonance can be tuned. The transmission spectra and the Poynting vector distribution of the THz sensor are simulated to analyze the properties of Fano resonance. Moreover, by establishing the coupled oscillator model (COM), the calculated Fano line-shapes are analyzed to illustrate the formation mechanism of Fano resonance. In addition, the high sensitivity and large FOM are also obtained, which makes the Au ring coupled bowtie BP structure a very promising platform for the research of high-performance THz sensors.

## 2. Results and Discussion

As we all know, BP has anisotropic properties and as demonstrated in Figure 1a, the atoms are arranged in a BP lattice in two directions: the armchair direction and the zigzag direction. The designed three-dimensional schematic of the unit cell of Fano sensor based on a bowtie BP coupled Au ring is shown in Figure 1b. The top view of the sensor is shown in Figure 1c. The unit cell is arranged in a periodic array with a square substrate with a substrate parameter *P* = 6 μm. The refractive index of the substrate is *n*. The Au ring can be fabricated by electron beam lithography (EBL) and inductively coupled plasma (ICP) dry etching processes. The mechanically stripped BP can be transferred to the substrate by a direct transfer method via polydimethylsiloxane. To analyze the influence of geometric dimensions on Fano resonance, we can adjust the inner radius (*R_1_*) and outer radius (*R_2_*) of the Au ring. What’s more, the side length (*a*) of the bowtie BP and the gap (*g*) between the neighboring bowtie structures are both fixed to be 1 μm. This small gap is to make the interaction between the neighboring bowtie structures even stronger. Because a larger gap will weaken the formation of dipoles, the surrounding electric field will be reduced, and the resonance will be weakened [38]. Moreover, it is worth noting that the interlayer interaction existing in the multilayer BP will cause the rearrangement of electrons between the layers, which may make the multilayer BP unstable in the practical application [39]. Therefore, the layer of bowtie BP is set as monolayer in this work. The thickness of the Au ring is set as 30 nm, which is about one skin depth at the operating frequencies [40]. In the simulation, the finite-difference time-domain (FDTD) method was used to analyze the transmission spectra of the Fano sensor [41]. The FDTD simulation with perfect matched layer is used to investigate the performances, and the grid sizes are set to be 10 nm × 10 nm. PML layers are set at the up and bottom boundaries of the structure, and the proposed structure is excited by a normal incident plane wave. The frequency-dependent complex relative permittivity of gold is characterized by the modified Debye-Drude dispersion model [40].

At THz frequencies, the permittivity of monolayer BP can be approximated by a Drude model as:(1)$$\varepsilon_{j}=\varepsilon_{\varepsilon }+\frac{i\sigma_{\alpha \beta }}{t\omega \varepsilon_{0}},$$

For a monolayer BP, the relative permittivity is given as $\varepsilon_{\varepsilon }=5.76$ [41] and the thickness *t* of BP is 1 nm. Vacuum dielectric constant is *ε_0_* and resonance frequency is ω. In Drude model, the conductivity $\sigma_{\alpha \beta }$ is given as:(2)$$\sigma_{\alpha \beta }=\frac{iD_{\alpha \beta }}{\pi \left(\omega +\frac{i\eta }{h}\right)}\;D_{\alpha \beta }=\frac{\pi e^{2}n}{m_{\alpha \beta }},$$
where αβ represents two lattice directions for BP and $D_{\alpha \beta }$ is the Drude weight. ℏ is the Planck constant, and *e* is the charge. We select the electron doping *n* = 5.5 × 10^12^ cm^−2^ and η is chosen to be 10 meV to describe the relaxation rate. $m_{\alpha \beta }$ is electron mass along with two lattice directions.

The transmission spectra of the proposed structure with and without bowtie BP are shown in Figure 2a. The insets in Figure 2a show the electric field distributions at the resonances of the three structures. The transmission spectrum of the bowtie BP coupled with Au ring shows the Fano resonance, while the transmission spectra of the Au ring and bowtie BP show the Lorenz-like resonance. It is demonstrated that the transmission spectrum of Au ring shows broad spectral characteristics, corresponding to the bright mode, while the transmission spectrum of bowtie BP shows narrower spectral characteristics due to the smaller scattering loss of BP, corresponding to dark mode. Compared with the electric field distribution of the Au ring and bowtie BP, the electric field distribution of the proposed hybrid structure (Au ring coupled with bowtie BP) is determined by the plasmon of the ring and the plasmon mode of the bowtie BP. The resonance at 25 μm arises due to interference between the bright plasmon modes of the Au ring and the dark plasmon modes of bowtie BP, resulting in an asymmetrical Fano line-shape. In order to explain the peak variation at the resonant wavelength in the Fano resonance transmission spectrum, the transmission phase of the coupled structure is shown in Figure 2b. The transmission phase reflects the sudden change in phase caused by the surface coupling structure. Compared to its corresponding transmission spectrum, it also shows a typical resonance spectrum. It can be seen that near the resonance wavelength, the phase change caused by the structure is π. In a two-terminal structure, this can be predicted based on time reversal symmetry, and also indicates that the abrupt change of the Fano resonance transmission spectra is due to the phase reversal [42].

To analyze the tunable property of the Fano resonance, we simulate the wavelength dependent amplitude transmission for various Fermi levels of BP and geometric parameters of bowtie BP. The modulation of the Fano resonance is attributed to the change in the refractive index properties of the BP caused by the Fermi level. According to the variation of the BP plasmon with the electron doping concentration, it can be obtained that the greater the electron doping concentration, the larger the Fermi energy level, and the smaller the resonance wavelength [43]. As illustrated in Figure 3, the simulated amplitude transmission spectra of the structure with Fermi level changes from 0.12 eV to 0.2 eV are demonstrated. The Fano resonance exhibits a blue shift as the BP Fermi level increases. The modulation of Fano resonances is attributed to the Fermi level induced change in the BP properties. In essence, the change of BP Fermi level is realized by changing the electron doping concentration, which will change the surface density of free carriers in the BP, thus leading to spectral deviation. There is a dependent relationship between the plasmon resonant wavelength $\lambda_{r}$ and the Fermi level $E_{F}$ in two dimensional electron gas [44]:(3)$$\lambda_{r}=\sqrt{\frac{2\pi^{2}\hbar c}{\alpha E_{F}\kappa_{sp}}},$$
where *a* is the fine structure constant, and $k_{sp}$ is the wave vector of surface plasmons. It can be seen from the expression that the Fermi level is inversely proportional to the resonance wavelength, that is to say, when the Fermi level increases, the resonance wavelength of the BP plasmon decreases. In addition, as illustrated in Figure 3b, the Fano resonance spectra corresponding to different bowtie BP geometric parameters are demonstrated. It can be seen that as the side length increases, the transmission peak redshift. This is because the increase in the size of the bowtie BP will redshift the BP plasmonic mode resonance, which will further result in the red shift of the transmission peak of the Fano resonance [45]. Besides, it can also be seen that as the side length of the bowtie BP increases, the line shape of the Fano resonance will change. Specifically, the larger the side length, the smaller the FWHM. To achieve better device performance, the bowtie BP side length should be selected as large as possible. The change of resonance wavelength of BP plasmon will further affect the Fano resonance. Therefore, we can dynamically control the transmission spectrum of BP nanostructures in a wide range by adjusting the Fermi energy and side length of Bowtie BP. This easily tuned Fano resonance can also be used as a mechanism for THz sensing.

To illustrate the influence of the geometrical parameters of the Au ring on Fano resonance, the transmission spectra of the bowtie BP coupled with the Au ring structure by varying the inner and outer radius are demonstrated in Figure 4a. A very clear Fano resonance line appears in the transmission spectrum. Unlike the Lorentz line, the Fano resonance spectrum is asymmetric. The slope of the left shoulder of the resonance peak is not the same as the right, which is a typical Fano line with a maximum and a minimum. As the inner radius and outer radius increase, the wavelength of the resonance peak redshifts, and the amplitude of the transmission peak becomes larger. Because the wavelength of the Fano resonance peak will be determined by the diameter of the ring, expressed as [46]:(4)λ=2Re(n)Lm−φ2π,
where *Re*(*n*) is the real part of the effective index, and *L* is diameter of the ring. Positive integer *m* is the number of antinodes of the plasmons wave and φ is the phase change. The increase in the inner and outer radius of Au ring will cause an increase in the diameter, which will cause the red-shift of the resonance wavelength. To further illustrate the dependence of Fano resonance on geometric parameters, in the transmission spectrum of Figure 4a, four points are selected to calculate the Poynting vector intensity, corresponding to the resonance peak and resonance dip under different inner and outer ring radii. By comparing the Poynting vector distributions from Figure 4b–e, it can be seen that the Poynting vector intensity at the resonance peak in Figure 4b is obviously much higher than the others. More importantly, the resonance in Figure 4b is caused by the coupling of the plasmon mode of the bowtie BP and the plasmon mode of the Au ring. On the contrary, a very weak coupling occurs between the Au ring and bowtie BP and has no plasmons coupled into the Au ring, as shown in Figure 4c,e. This means that the Fano resonance arises from the interaction between the plasmon mode of the Au ring and the bowtie BP. Furthermore, it is worth noting that there is an interesting phenomenon in Figure 4a. As the inner and outer radius increases, the slopes of the left and right sides of the resonance peak tend to be the same, that is to say, it shows a Lorentz line. This can be explained by the Poynting vector distribution in Figure 4b,d. From Figure 4b, it can be seen that the plasmon mode excited by bowtie BP and the Au ring are coupled with each other to produce Fano resonance. On the contrary, in Figure 4d, only the plasmon mode of bowtie BP is excited, which does not constitute a condition for exciting Fano resonance, thus producing a symmetric Lorentz resonance line. Therefore, when the inner and outer radius of the ring is 2.5 μm and 3 μm, it is the optimal parameter to obtain Fano resonance.

To explain the mechanism of the formation of Fano resonance, the proposed structure can be simplified as an energy model to describe Fano resonance. As shown in Figure 5a, the excitation of Fano resonance is illustrated. The bright mode has a large dipole moment and can be directly excited by incident light, while the dark mode with almost negligible dipole moment cannot be directly excited by incident light [11,47]. However, the dark mode can be excited by the near field related to the bright mode, and coupled with the bright mode to excite Fano resonance. In fact, Fano resonance is the result of competition between bright mode (Au ring plasmon) and dark mode (bowtie BP plasmon), and its linear shape depends on the coupling between the relative dipoles [35]. Therefore, we can get a Fano parameter *q* from Fano theory to describe the relative dipole strength [48]:(5)$$\;q=\frac{1}{\pi E_{pl}g}\times \frac{D_{dark}}{D_{bright}},$$
where $E_{pl}$ is plasmon electromagnetic density and *g* is the coupling strength. The total dipole moment of bowtie BP plasmon dark mode is $D_{dark}$ and $D_{bright}$ is the dipole moment of Au ring plasmon bright mode. Generally, the bright plasmon mode has a large dipole moment, while the dark plasmon mode has zero or negligible dipole moment [11], which will cause the absolute value of *q* to be much smaller than 1. In this case, the asymmetrical Fano line rarely appears. However, if the dipole moment of the dark plasmon mode is almost the same as the dipole moment of the bright plasmon mode, that is, the absolute value of *q* is approximately equal to 1, a typical asymmetric Fano line will be obtained.

As illustrated in Figure 5b, natural line shapes for different values of *q* are demonstrated. When |*q*| = 0.1, the shape of the spectral line is symmetrical, which is very close to the Lorentz line-shape. As |*q*| increases to 1, the shape of the spectral line gradually changes until it becomes a very typical asymmetric Fano line-shape. So, the closer the absolute value of *q* is to 1, the more standard the sharp asymmetric Fano line-shape will be.

What is more, in order to gain a qualitative understanding of Fano resonance, the bowtie BP coupled Au ring system is analyzed based on the coupled oscillator model (COM) [49]. According to the energy-conservation and time-reversal symmetry, the theoretical model can be stated as:(6)$$T=\frac{{\left[\left(\omega_{0}-\omega \right)t\pm r\gamma \right]}^{2}}{{\left(\omega_{0}-\omega \right)}^{2}+\gamma^{2}},$$
where $\omega_{0}$ is the resonance center frequency and γ is the total loss rate of the system. In our transmission simulation results of the coupled structure in the case of normal incidence, the center frequency of the resonance is 7.5 THz. In particular, the real constants *r* and *t* follow the relation *r*^2^ + *t*^2^ = 1. The fitting result is demonstrated in Figure 5c. As it can be seen from the black line, this is a very typical Fano resonance line. By comparing the simulation result with the fitting result, we find that the amplitude of Fano resonance is roughly the same, but the resonance line width of the fitting result is much smaller than that of the simulation result. On the one hand, the radiation energy of BP is high, that is, the energy dissipation rate is high, which will lead to the energy loss in the process of light matter interaction [50]. It is worth noting that for lossy systems, the loss has a great effect on the resonance characteristics. When the loss rate γ is greater than the central resonance frequency $\omega_{0}$, the system cannot resonate due to excessive loss. For the case of γ << $\omega_{0}$, the spectrum will exhibit sharp Fano line-shape. If γ of the system is neither much greater than nor much less than $\omega_{0}$, the spectrum exhibits a smooth Fano line-shape. Therefore, in our system, because of the loss caused by the large radiation energy of BP, the Fano line presents smooth changes rather than sharp curves. On the other hand, this can be attributed to the fact that the absolute value of Fano parameter *q* based on the BP excitation subsystem cannot reach 1. Generally, in the coupled system, the dipole moment of bright modes is greater than that of dark mode. Therefore, the absolute value of the Fano parameter of our structure cannot be infinitely close to 1, but it is much larger than 0, and finally shows a comparatively wider sharp Fano resonance line.

In order to further clarify the influence of the relationship between Fano parameter *q* and dipole moment on Fano line-shape, the simulated surface charge distributions at the resonance peak with different inner and outer radii are plotted in Figure 6. For normal incidence, most of the surface charges concentrate at the edges of the structure. As can be seen from Figure 6a,b, this is a typical dipole ring plasmon resonance, and the same types of charges are accumulated together. Therefore, in this resonance mode, the resonance spectrum of this mode will not change drastically. The resonant peak can be excited by the plane wave, indicating the bright mode. We know that the magnitude of the electric dipole moment is proportional to the amount of charge and the distance between the charges [51]. Therefore, with the decrease in the inner and outer radius, the total dipole moment of the Au ring plasmon mode decreases, leading to the increase in absolute value of Fano parameter *q*. This further proves that when the inner and outer radius of the ring is 2.5 μm and 3 μm, the Fano resonance is more similar to the standard sharp asymmetry line-shape, with a narrower line width. In addition, compared with the charge distribution in Figure 6b, the charge distribution in Figure 6c is asymmetric. It is worth noting that the surface charge distribution of bowtie BP presents a quadrupole near field pattern and is not easily excited directly by plane waves, indicating the dark mode. For Fano resonance, a large number of opposite charges are concentrated on the coupling structure, and the strong electric fields are confined to the structure, which will improve the sensitivity of the whole structure [52]. Therefore, in the Fano resonant mode, drastic changes in the modal resonance spectrum will be caused. This further indicates that the formation of Fano resonance is caused by the dipole plasmon bright mode of the Au ring coupled with the quadrupole plasmon dark mode of the bowtie BP.

In order to investigate the sensitivity of the Fano resonance of the sensor to the refractive index *n* of the substrate, the refractive index of the substrate was changed. As shown in Figure 7a, the transmission spectra of the coupled structure by varying refractive index *n* of 3.0–4.5 RIU in increments of 0.5 RIU are demonstrated. The inner and outer radius of the ring is set as to 2.5 μm and 3.0 μm. The Fano resonance peak showed a redshift as *n* increased. This result is also consistent with the trend predicted by the Fano resonance wavelength formula (4). Besides, the results in Figure 7a indicate that the wavelength shifts from the peaks to the dips are about 6.2 μm, 6.2 μm, 7 μm and 7.5 μm, respectively. This long wavelength variation can provide the structure with high sensitivity spectral response to the refractive index variation in a wide range of media. Figure 7b shows the shift of the Fano resonance peak as a function of the refractive index change $\delta_{n}$. The functional relationship is almost linearly fitting. In addition, the sensitivity of the sensor can be calculated based on the linear result in Figure 7b. The sensitivity *S* can be expressed as: *S* = (δλδn), which means the transmittance change induced by the change of refractive index [31]. It can be understood as the amount of the red shift or blue shift of the formant when the refractive index change unit of the substrate is 1. The sensitivity of the sensor is 8.1 μm/RIU, 8.5 μm/RIU and 9.3 μm/RIU according to Figure 7b when *R_1_* − *R_2_* = 3.0 μm − 2.5 μm, *R_1_* − *R_2_* = 3.5 μm − 3.0 μm, and *R_1_* − *R_2_* = 4.0 μm − 3.5 μm, respectively. It can be seen that the sensitivity increases with increasing radius. This fully demonstrates that the sensitivity of the proposed Fano sensor can be adjusted by tuning the device structural geometric parameters. The highest sensitivity can be obtained when the inner and outer radius is 4.0 μm and 3.5 μm. Figure 7c gives a more detailed analysis on the wavelength variations with respect to the refractive index of the substrate. It can be seen that as the refractive index increases, the resonance peak and dip both red-shift. Obviously, the wavelength has a linear relationship with n. Moreover, FOM as a key factor is widely used to evaluate the performance of the Fano resonance. FOM can be defined as: FOM = S/FWHM [45], where FWHM is the full width at half-maximum of the resonance. It is predicted that in order to obtain a larger FOM, ultra-small line width and high sensitivity are preferred. Therefore, from Figure 7c, the FOM values can be calculated as 69.3, 66.6 and 65.5 when *R_1_* − *R_2_* = 3.0 μm − 2.5 μm, *R_1_* − *R_2_* = 3.5 μm − 3.0 μm, and *R_1_* − *R_2_* = 4.0 μm −3.5 μm, respectively. It can be found that as the radius increases, the FOM value decreases. It is due to the fact that as the radius increases, the linewidth of the Fano resonance increases. It can be seen from Figure 4a, as the inner and outer radius increases, the FWHM of the Fano resonance also increases. Therefore, although the sensitivity increases as the radius of the ring increases, the FWHM increases further. The highest FOM can be obtained when the inner and outer radius is 2.5 μm and 3 μm. This FOM value is significantly greater than that in the previous reports [26,45]. For example, a refractive index sensor based on metal-insulator-metal waveguides coupled double rectangular cavities was proposed to obtain the FOM value of 7.5, which is far lower than our results [45]. Therefore, the high sensitivity and FOM value of the device make the proposed structure spawn a wide range of applications in the field of plasmon nano-sensors at THz frequency.

## 3. Conclusions

In summary, Fano resonance has been demonstrated in our proposed coupled structure, which consists of bowtie BP and Au ring. The strength and resonance frequency of Fano resonance can be dynamically adjusted by changing the geometric parameters of the Au ring and the Fermi energy level of BP at THz frequency, respectively. In addition, Poynting vector distributions are simulated to illustrate the dependence of Fano resonance on geometric parameters. What is more, by establishing the COM, the fitting resonance curves of the coupled structure well match with the simulated resonance results. Moreover, numerical results show that a high frequency sensitivity of 9.3 μm/RIU and a high figure of merit (FOM) of 69.3 can be obtained of our proposed structure, which has wide ranging potential applications for plasmon resonance sensors at THz frequency.

## Figures and Tables

**Figure 1 nanomaterials-11-01442-f001:**
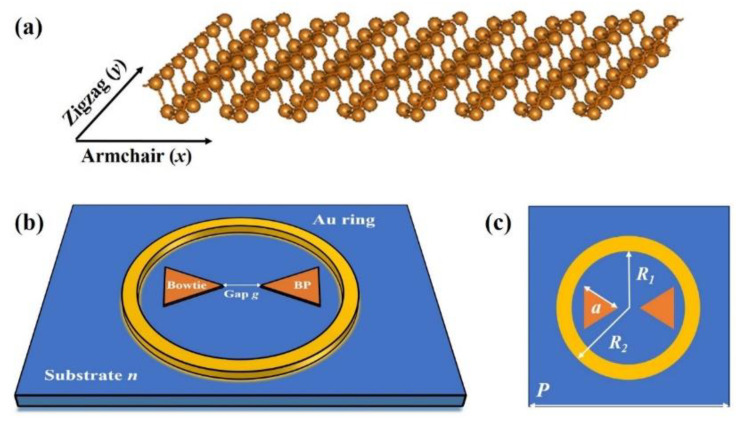
(**a**) Schematic diagram of BP crystal structure. (**b**) The three-dimensional schematic of Fano sensor based on BP bowtie coupled with Au ring structure. The gap distance between the neighboring bowtie BP structures is *g*. (**c**) The top view of bowtie BP coupled with Au ring structure. The inner radius of the ring is *R_1_* and outer radius is *R_2_*, and the side length of the BP bowtie is *a*.

**Figure 2 nanomaterials-11-01442-f002:**
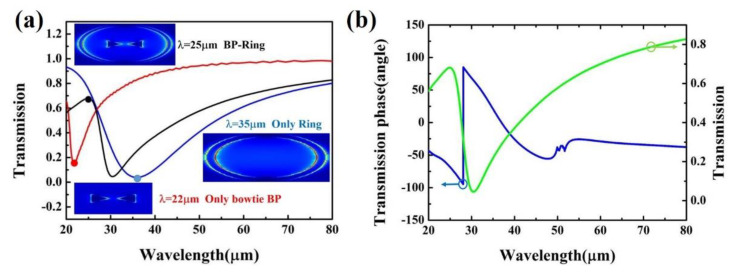
(**a**) The transmission spectra through the proposed structure composed of Au ring only (blue), bowtie BP (red) and both the Au ring and the bowtie BP (black). Insets: electric field intensity distributions. (**b**) Transmission phase (blue) of the proposed structure. Transmission spectra (green) is the transmission spectra of the proposed structure composed of both Au ring and bowtie BP in figure a. Please note that in the electric field distribution of the inset, the ring looks like an elliptical ring, which is caused by the difference in the step length of the X-axis and the Y-axis.

**Figure 3 nanomaterials-11-01442-f003:**
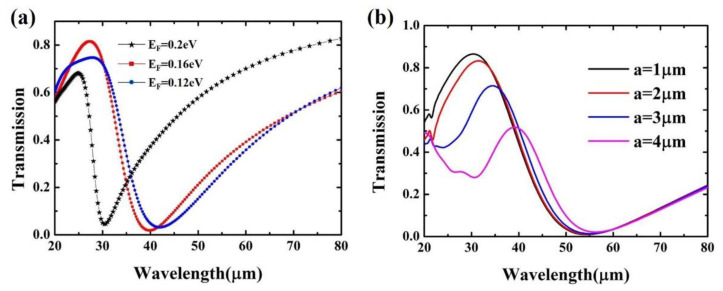
(**a**) Simulated amplitude transmission spectra of the structure with Fermi level changes from 0.12 eV to 0.2 eV. (**b**) Transmission spectra of the BP bowtie coupled with Au ring structure as a function of the side length of the bowtie BP. The side length ranges from 1 μm to 4 μm.

**Figure 4 nanomaterials-11-01442-f004:**
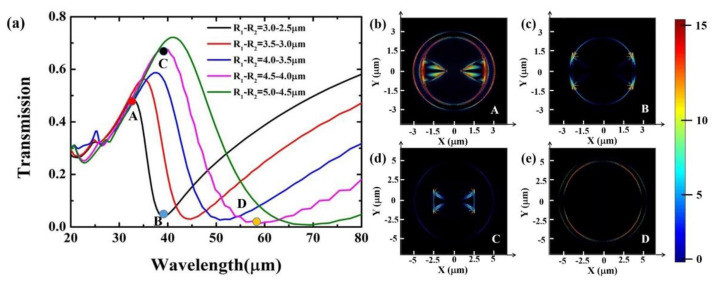
(**a**) Transmission spectra of the BP bowtie coupled with Au ring structure as a function of the inner and outer radius of the ring. The inner radius of Au ring ranges from 2.5 μm to 4.5 μm, and the outer radius of Au ring ranges from 3 μm to 5 μm. The Poynting vector distribution of the coupled structure in the X–Y plane at a wavelength of (**b**) 32.5 μm (resonance peak) and (**c**) 38.82 μm (resonance dip) when the inner and outer radius of the ring is 2.5 μm and 3 μm. The Poynting vector distribution of the coupled structure in the X–Y plane at a wavelength of (**d**) 38.82 μm (resonance peak) and (**e**) 58.67 μm (resonance dip) when the inner and outer radius of the ring is 4 and 4.5 μm.

**Figure 5 nanomaterials-11-01442-f005:**
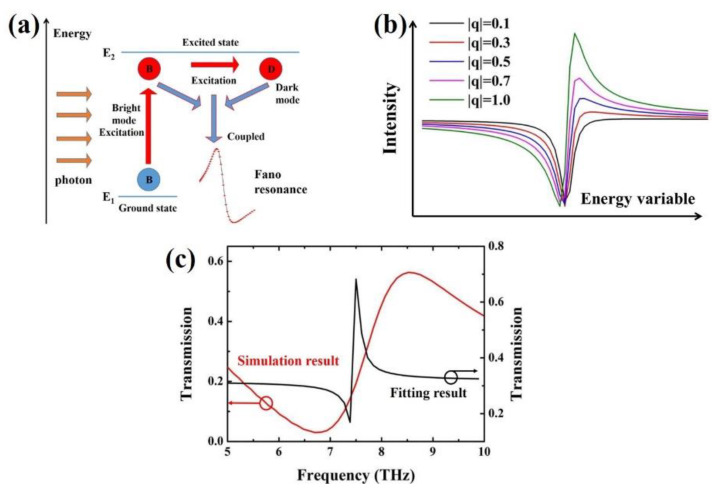
(**a**) Schematic representation of the plasmon hybridization and the associated interaction between the bright and dark modes. (**b**) Natural line shapes for different values of *q*. (**c**) Transmission spectral for the fitting result of coupled-mode theory of the coupled structure in the case of normal incidence.

**Figure 6 nanomaterials-11-01442-f006:**
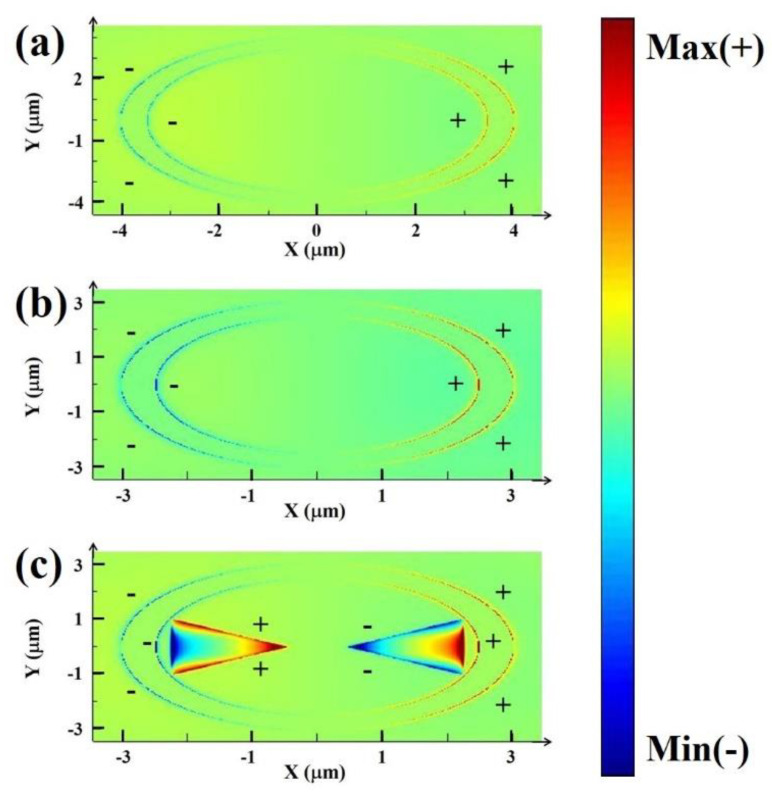
The simulated surface charge density distributions at the resonance peak with only the Au ring in the X–Y plane. (**a**) The inner and outer radius of the ring is set as to 3.5 μm and 4.0 μm. (**b**) The inner and outer radius of the ring is set as to 2.5 μm and 3.0 μm. (**c**) The simulated surface charge density distributions at the resonance with the proposed coupled structure in the X-Y plane. The inner and outer radius of the ring is set as to 2.5 μm and 3.0 μm. Please note that in the surface charge density distributions, the ring looks like an elliptical ring, which is caused by the difference in the step length of the X-axis and the Y-axis.

**Figure 7 nanomaterials-11-01442-f007:**
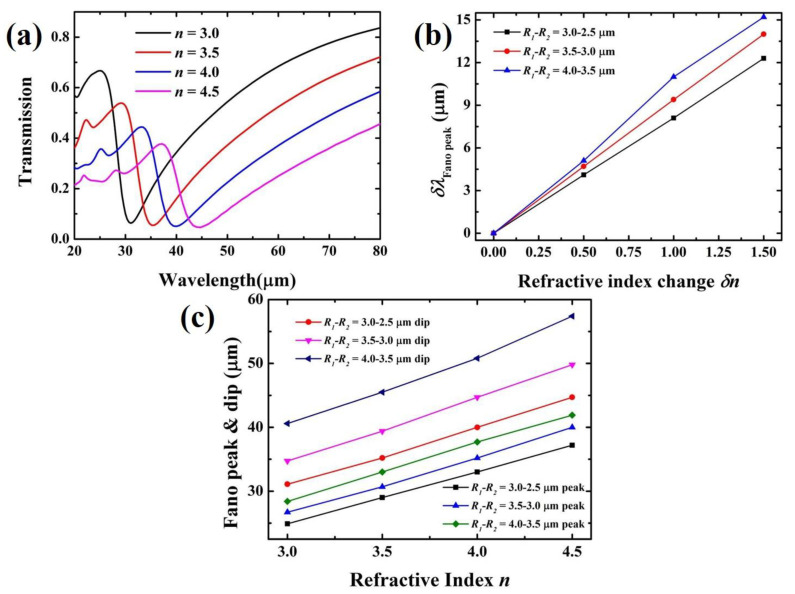
(**a**) Transmission spectra of the bowtie BP coupled with Au ring structure as a function of the refractive index of the substrate. The inner and outer radius of the ring is set as to 2.5 μm and 3.0 μm. (**b**) The shift of the Fano resonance peak as a function of the refractive index change $\delta_{n}$. (**c**) The peak-wavelength and dip-wavelength variations with the refractive index of the substrate.

## Data Availability

The study did not report any data.

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
