# Peer review of "Actively Tunable Fano Resonance Based on a Bowtie-Shaped Black Phosphorus Terahertz Sensor"

_nanomaterials, 2021, doi:10.3390/nano11061442_

Round 1

Reviewer 1 Report

ring-bowtie structure is shown for sensor or filter application in THz spectral band. physics of performance is nicely described. for improvements, please define all parameters used in formula 2. incidencein not vertical but normal, please correct. all pictures are horizontally stretched. this makes loss of message on geometry: ring becomes ellipsis. please make x-y scales equal. 

BP is a tricky material to handle. please explain what is expected in experimental implementation. how we can make one monolayer, preparation is usually used: solution casting, etc?  THz band has a lot of water absotption bands. how this will affect real performance of suggested memtasurfaces when water is attached to BP or metal. 

in IR-THz spectral range bow-ties of different geometry and structure are used to tailor spectral sharpness. Please discuss how Fano resonance is changing if spectral width of bowtie is engineered by its structure (fractal or other pattern). 

Polarization of bowtie defines spectral performance. then BP has also anisotropy. how  they talk to each other?

Reviewer 2 Report

A Drude classical model was used to predict Fano resonance in a specific thin film structure.
It is well known that the permittivity of thin films depends on the film thickness.
The authors should justify that the above Drude model is applicable for THz waves and for a 10 nm thin film thickness.

The manuscript is not self-contained which means one should read other sources, in order to understand the theory.

At thin films, a crystallografic orientation of them become a significant factor. However, neither BP, no gold films were not characterised for the crystallography.

For this manuscript to be publishable, at least one experimental comparison for an intermediate or a final variable should be provided.

Author Response

Plaese see the attachment.

Round 2

Reviewer 1 Report

as per previous review: insets in fig 2 and fig 6 is misleading due to elliptical shape. 
